# An Ultra-Broadband Design of TM-Pass/TE-Stop Polarizer Based on Multistage Bragg Gratings

Yue Dong *, Yu Liu, Yin Xu and Bo Zhang

Department of Electronics, School of Internet of Things, Jiangnan University, Wuxi 214122, China; 6191916024@stu.jiangnan.edu.cn (Y.L.); yin.xu@jiangnan.edu.cn (Y.X.); zhangb2018@jiangnan.edu.cn (B.Z.)
* Correspondence: y.dong@jiangnan.edu.cn; Tel.: +86-137-7095-3767

**Abstract:** In this paper, a multistage Bragg grating with various kinds of periods is introduced in the design of a reflection-based TM-pass/TE-stop polarizer. The cascade grating sections reflect a wide wavelength range of the TE polarization state. Additionally, on the other hand, the TM polarization state always passes through the waveguide. Such a design facilitates the polarizer working bandwidth, which is defined as the wavelength range with an extinction ratio of greater than 20 dB, and can reach 231 nm using only three grating sections. Meanwhile, the incision loss is always less than 0.42 dB over the working wavelength band. Furthermore, if a slightly higher loss is permitted, the polarizer working bandwidth can be extended to further than 310 nm using five grating sections.

**Keywords:** polarizer; Bragg gratings; integrated photonics

## 1. Introduction

In photonics integrated circuits (PICs), the performances of polarization sensitive devices and systems are always limited to large birefringence [1]. In this case, there is a great demand to produce a polarization management device to perform polarization states manipulation in PICs. To date, a number of device structures have been reported to perform various polarization control functions, including polarization beam splitters (PBS) [2–4], polarization splitter-rotators (PSR) [5–7], and polarizers [8–10]. Theoretically, the PBS and PSR require restrict phase matching conditions to realize splitting operations so their structural geometries have to be carefully controlled in design and fabrication. In contrast, the polarizer is a much simpler but effective device that eliminates the undesired polarization state by means of leakage, absorption, or reflection.

A high-performance polarizer should have a high extinction ratio (ER), low insertion loss (IL), and broadband operations. To achieve such goals, leakage-mode-based polarizers were initially realized by using a shallowly etched ridge waveguide. Due to the low birefringence nature of such a waveguide, the polarizers usually have a fairly long footprint that is greater than 100 μm [11]. Additionally, in some particular case, the device footprint is greater than 1 mm [12]. In the next a few years, the slot-assisted [12] and subwavelength gratings (SWG) assisted [13] waveguides will be introduced in leakage-mode-based polarizers to achieve broadband (>100 nm), high ER (>30 dB), and low IL (~0.5 dB) operations simultaneously. Although the device footprint was dramatically decreased to several tens of microns, it still has the potential to further reduce the device length for higher integration density. In recent years, the hybrid plasmonic gratings (HPG) [14–16] have offered another method by which to reach a high ER (>20 dB) and broadband (>100 nm) operation with, impressively, a shorter device footprint. In particular, with a nanoscale plasmonic waveguide, only an ultracompact device length of 0.63 μm is required for a working bandwidth of 100 nm [16]. However, even such kind of short device footprint is able to realize a bandwidth greater than 230 nm [14], and the considerable light wave absorptions lead HPG to suffer from ultra-high loss of greater than 2 dB.

The reflection-based polarizer provides a relatively good trade-off between the device footprint and loss. Typically, it reflects the unwanted polarization state with Bragg gratings at a reasonable length of slightly less than 10 μm [10,17]. Additionally, the achieved IL are always less than 0.5 dB over the entire operating wavelength band. However, in terms of the working bandwidth, it is quite limited due to the narrow reflection wavelength band for a certain period of Bragg gratings.

In this paper, an ultra-broadband TM-pass/TE-stop polarizer is proposed for a silicon-on-insulator (SOI) platform. By using multistage Bragg grating sections, the reflection wavelength region for the TE polarization state is effectively extended. Hence, the bandwidth defined as the wavelength region with an ER of greater than 20 dB is extended as well. Although the changed effective index between the nonboring grating sections slightly increases the reflection power of the TM polarization state, the IL is always maintained at an acceptable level. After optimizing device geometries, the proposed polarizer achieves its highest ER at around 1550 nm by using three grating sections with three distinct periods of 320 nm, 340 nm, and 360 nm. Additionally, the device working bandwidth and IL are 231 nm and 0.42 dB, respectively. If a slightly higher loss in the TM polarization state is permitted, the polarizer bandwidth can be extended to 310 nm using five grating sections.

## 2. Device Structure and Principles

Figure 1 shows the three-dimensional (3D) schematic diagram of the proposed TM-pass polarizer. The device was designed using a silicon-on-insulator (SOI) platform with a 220 nm thick silicon wafer. In addition, it also has a 3 μm thick buried oxide layer and 3 μm thick silicon dioxide ($SiO_2$) up claddings. The device's main body is composed of multistage grating sections and each section has an independent period. According to the Bragg condition, the gratings reflect a certain wavelength of incident light wave that is determined by the grating period and the corresponding effective index. In the proposed polarizer, the TM polarization sate normally has a much lower effective index than the TE polarization state. Therefore, for fixed grating structure geometries, when the incident TE polarization state meets the Bragg condition and is reflected, the TM polarization state transmits in a subwavelength grating (SWG) waveguide. The proposed polarizer has multiple grating sections and each kind of grating period reflects a certain wavelength range of the TE polarization state. Additionally, the overall reflected spectra are expanded compared with that when using only one kind of grating period. Then, it leads to a much broader operating bandwidth. Initially, the polarizer is designed to have a universal width of $W_1$ and all grating sections also have another universal width of $W_2$. Additionally, all gratings are set to have a constant period of $\Lambda$ = 370 nm and their duty cycle is set to a constant of a/$\Lambda$ = 0.5 as well.

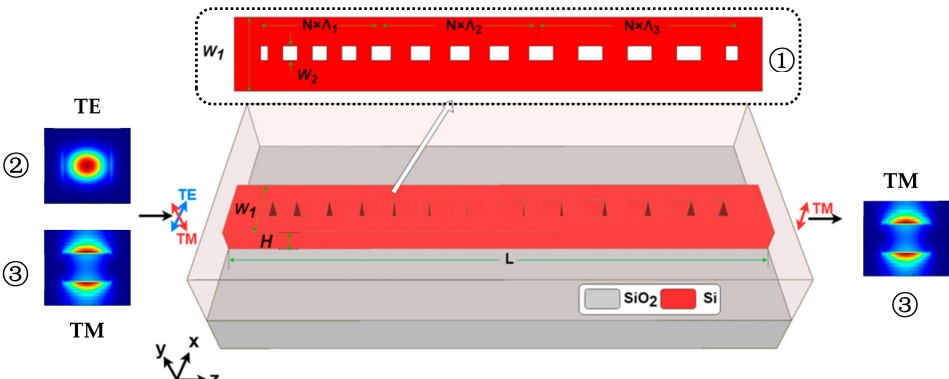

**Figure 1.** Schematic diagram of the proposed polarizer. Inset ①: Top view of the proposed TM—pass polarizer; ②: Mode profile of TE polarization state; ③: Mode profile of TM polarization state.

In order to have understand the reflection wavelength band of TE polarization state, we swept the normalized wavenumber $k_{norm}$ in the propagation direction from 0.3 to 0.5 to

perform band structure analysis [18,19] for both TE and TM polarization states. Then, the corresponding band diagrams were extracted as shown in Figure 2a. Apparently, due to the Bragg reflection, the TE polarization state has a wide wavelength gap of between 1410 nm to 1690 nm in the band diagram. In contrast, the TM polarization state is supported at all times across the entire wavelength range from 1100 nm to 1800 nm that also covers the wavelength gap of the TE polarization state. This is because the short device height only generates quite limited effective indices for the TM polarization state that never meet the Bragg condition in the target wavelength region. It implies that the TM polarization state will always pass through the proposed device in such a wavelength range. To quantitatively evaluate the device transmission behavior, the 3D-FDTD calculations were performed to separately characterize the transmittance and reflectance for both the TM and TE polarization states. As shown in Figure 2b, the maximum transmittance of the TM polarization state is about −0.42 dB, which is equivalent to slightly more than 90% of the TM power being able to pass through the device. Regarding to TE polarization state, it has very low transmittances of less than −20 dB over the above-mentioned wavelength gap, i.e., 1410 nm to 1690 nm. Although such a transmission performance seems acceptable in the communication band, we optimized the device geometries so that a broader working bandwidth could be achieved for the proposed polarizer.

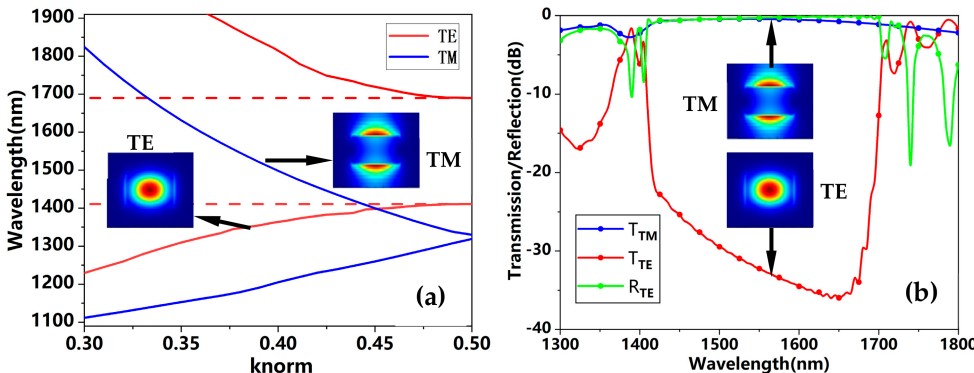

**Figure 2.** (**a**) The band structure diagram for both TE and TM polarization states; (**b**) The wavelength dependent transmittance and reflectance of TE and TM polarization states under the initialized device geometries. Inset: Mode profiles of TE and TM polarization states.

In general, the device structure is equivalent to a slot waveguide embedded with Bragg gratings, as the slot width $W_2$ normally has a greater impact on the loss of light wave propagating in the waveguide. Thus, we simply swept the slot width $W_2$ from 100 nm to 300 nm and performed 3D-FDTD simulations to evaluate the wavelength-dependent transmittances. As demonstrated in Figure 3a, the transmittances increased dramatically as the slot width decreased. This is because the decreased slot width led to a larger effective index of the TM polarization state so that it was better confined in the waveguide with a lower insertion loss. However, when the highest TM transmittance was reached at −0.2 dB with a slot width of $W_2$ = 100 nm, the wavelength band for the reflected TE polarization state decreased to only 180 nm, that is, from 1460 nm to 1640 nm. The much narrower reflection wavelength band results in a worse working bandwidth. Nevertheless, the additional grating sections are able to extend the reflected TE wavelength band but generate more loss. We selected the slot width $W_2$ as 100 nm to ensure an acceptable low loss operation in the cascaded grating sections.

Apart from the slot/grating width $W_2$, the overall waveguide width $W_1$ also greatly impacts the device transmission performance. We kept the slot width $W_2$ = 100 nm and swept $W_1$ to evaluate the wavelength-dependent transmittances for TE and TM polarization states separately. In Figure 4a, the highest transmittance for TM polarization state is shown to be always greater than −0.2 dB when $W_1$ is larger than 500 nm. Additionally, the center wavelength of the transmitted TM polarization state is almost unchanged. Regarding the

TE polarization state, as shown in Figure 4b, since the TE effective index increases as the $W_1$ increases, the reflection is red shifted towards a longer wavelength region according to the Bragg condition. In order to match the working wavelength region for both TM pass and TE reflection operation, $W_1$ was selected as 600 nm.

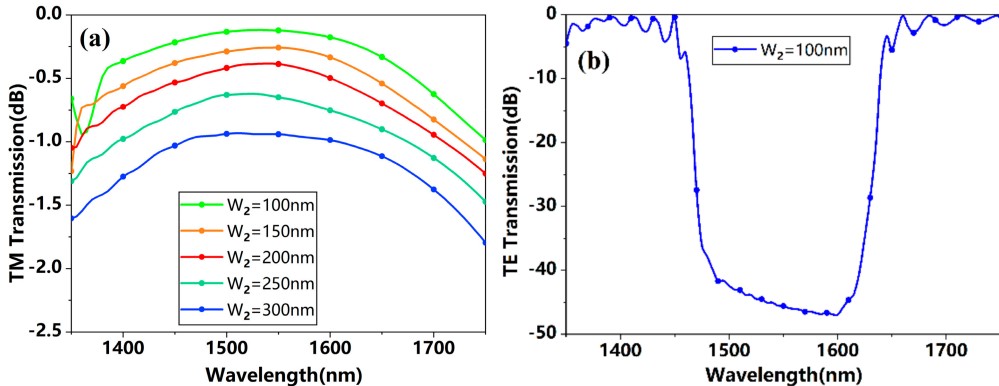

**Figure 3.** (**a**) The wavelength dependent transmittance of TM polarization state under various different slot width $W_2$ from 100 nm to 300 nm; (**b**) The wavelength dependent transmittance of TE polarization state under $W_2$ = 100 nm.

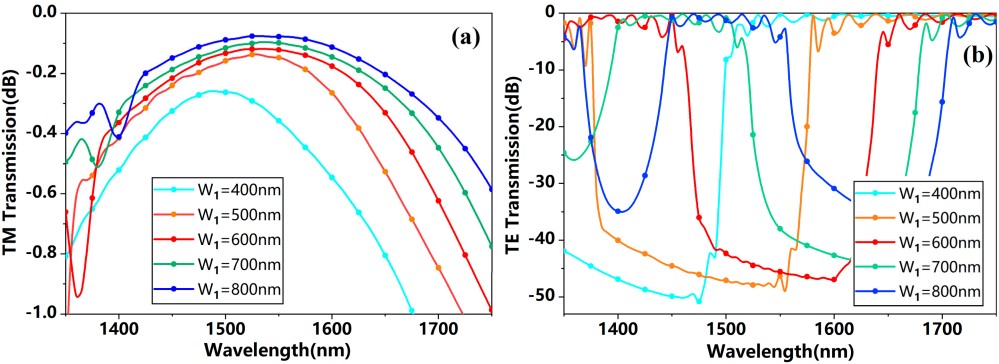

**Figure 4.** The wavelength dependent transmittances evaluated with various waveguide widths $W_1$ for (**a**) TM polarization state and (**b**) TE polarization state.

According to the Bragg condition, i.e., $\Lambda_{Bragg} = \lambda/(2n_{eff})$, the Bragg period is proportional to the working wavelength and inversely proportional to the corresponding effective index. Conversely, the reflection wavelength is dependent on the grating period. When gratings with various different periods are introduced in the proposed polarizer, the overall working wavelength region is ideally extended as the superposition of all the separated reflected wavelength regions. In order to compensate for the shrunken reflection wavelength region for TE polarization states, due to the decreased slot width $W_2$, two more grating sections were added in the proposed polarizer to extend the wavelength region of TE reflection and then the polarizer working bandwidth was also extended.

As demonstrated in Table 1, the grating period increased from 300 nm to 390 nm with an increment of 10 nm to characterize their corresponding wavelength regions for TE reflection. Apparently, as the period increased, the region of the reflected was red shifted and the corresponding bandwidth, which is defined as the wavelength range that has an ER greater than 20 dB, increased from 150 nm to 180 nm. In particular, if the grating period $\Lambda$ = 340 nm, the Bragg reflection of the TE polarization state becomes centered at 1550 nm with a bandwidth of 165 nm. Meanwhile, approximately 70% of the wavelength region overlaps for TE reflection generated by $\Lambda$ = 340 nm and its neighboring grating periods, i.e., $\Lambda$ = 330 nm and $\Lambda$ = 350 nm. Therefore, in order to design a polarizer centered at 1550 nm and with a bandwidth as broad as possible, alternating periods of $\Lambda$ = 340 nm,

namely Λ = 320 nm and Λ = 360 nm, were selected instead of neighboring periods for the other two grating sections to reduce the wavelength overlap as much as possible.

**Table 1.** TE reflection wavelength region and the bandwidth of the proposed polarizer.

| Grating Period Λ (nm) | 300 | 310 | 320 | 330 | 340 |
|---|---|---|---|---|---|
| Reflection Wavelength Region (nm) | 1365~1515 | 1395~1540 | 1420~1575 | 1445~1605 | 1470~1635 |
| Bandwidth (nm) | 150 | 145 | 155 | 160 | 165 |
| Grating Periods Λ (nm) | 350 | 360 | 370 | 380 | 390 |
| Reflection Wavelength Region (nm) | 1495~1660 | 1520~1690 | 1545~1715 | 1565~1745 | 1590~1770 |
| Bandwidth (nm) | 165 | 170 | 170 | 180 | 180 |

### 3. Results and Discussions

#### 3.1. Polarizer Performances

Under the optimized device geometries mentioned above, we conducted full 3D-FDTD simulations for TE and TM polarization states, respectively. Figure 5a shows the power propagation of the TM polarization state at 1550 nm. Clearly, the TM polarization state was perfectly confined in the proposed polarizer with minimal loss. For the TE polarization state, Figure 5b–d demonstrates its power propagation at 1500 nm, 1550 nm, and 1600 nm, respectively. Since the most left grating section has a period of 320 nm, according to Table 1, the TE polarization state at 1500 nm is firstly reflected at this stage. However, at 1600 nm, most of the optical power of the TE polarization state is able to pass through the grating section with a period of Λ = 320 nm. Additionally, the Bragg reflection begins in the 340 nm grating section. Such operations meet the grating period-dependent reflection analysis in Table 1.

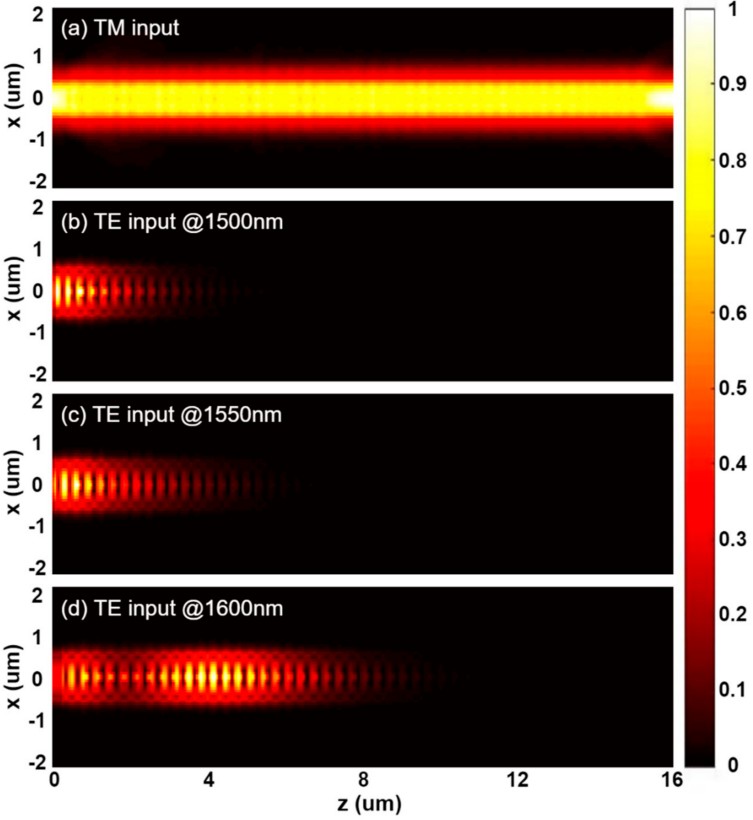

**Figure 5.** Power propagation of (**a**) TM polarization state at 1550 nm; (**b**) TE polarization state at 1500 nm; (**c**) TE polarization state at 1550 nm; (**d**) TE polarization state at 1600 nm.

Figure 6a shows the wavelength-dependent transmittances under the grating periods of $\Lambda_1 = 320$ nm, $\Lambda_2 = 340$ nm, $\Lambda_3 = 360$ nm, independently, and the superposition of the three kinds of grating periods. It is clear that the reflection wavelength band has a red shift as the period increases. Additionally, the neighbouring periods have an overlapped wavelength band and each period has its unique wavelength band. However, the final transmission spectrum by using multiple grating sections is not simply the superposition of the three independent wavelength bands. The reflectance in the overlapped wavelength band strengthened owing to the multiple reflections in all grating sections, but the remaining wavelength band weakened because it was only reflected in one of the sections. Nevertheless, the reflection wavelength range under the three grating sections has much broader operations than when using one single grating section. Taking account of the negative impact of reflections on the light source, it is better to have a non-reciprocal device in front of the polarizer, such as a circulator [20] or isolator [21]. In Figure 6b, the ER and IL are determined using the simulated TE and TM transmittances under three grating sections. At 1550 nm, the extinction ratio and insertion loss are 51.6 dB and 0.09 dB, respectively. Additionally, under a device length of 15.3 μm, the polarizer achieves a relatively high ER of greater than 20 dB over the wavelength region from 1434 nm to 1665 nm. Meanwhile, the IL is always less than 0.42 dB within this wavelength region.

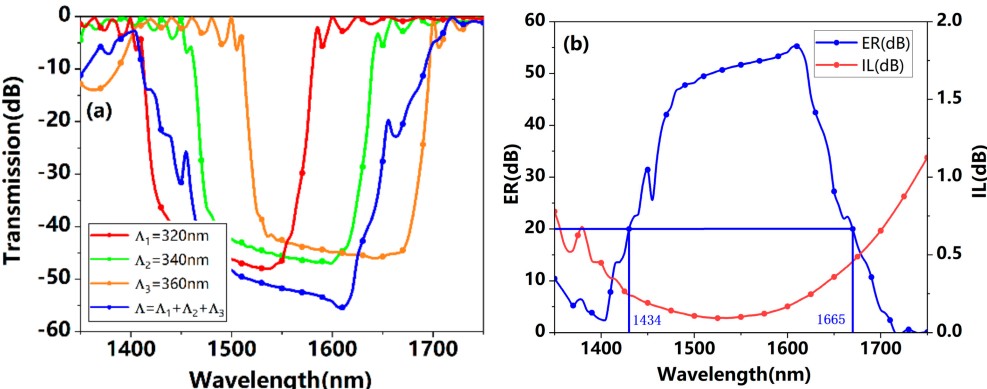

**Figure 6.** (**a**) Transmittances under gratings periods of 320 nm (red), 340 nm (green), 360 nm (brown), and the superposition of the three periods (blue); (**b**) The extinction ratio and insertion loss of the proposed polarizer with a length of 15.3 μm.

As a rule of thumb, the length of the polarizer has a considerable impact on the polarizer performance, such as ER and IL. With the assumption that the three grating sections have the same number of periods N, we tested the number of periods N from 8 to 15 and evaluated the corresponding ER and IL. As demonstrated in Figure 7a, the IL at 1550 nm for the TM polarization state was insensitive to the device length, and remained consistently around 0.1 dB. Over the entire wavelength region, the highest IL slightly increased from 0.3 dB to 0.4 dB, as shown in Figure 7b. In terms of ER, there was a significant increase along with the increased number of periods N, owing to the contribution of grater reflected TE power. In this case, the bandwidth increased along with the utilized number of periods. According Figure 7b, if the utilized number of periods increased from 8 to 15, the bandwidth also increased from 180 nm to 231 nm. However, when the number of periods N was greater than 12, the rise in the bandwidth rate slowed down dramatically. Hence, N = 12 provides a more compact solution (~12.2 μm) if a bit narrower bandwidth of 225 nm is achieved.

Apart from the above discussed device geometries, we also conducted a device performance analysis on device height h and grating duty cycle a/$\Lambda$. When the device height was set as h = 200 nm, the TM effective index decreased. Especially for the wavelength region longer than 1600 nm, the TM effective index was evaluated to be less than 1.6 and was too close to the SiO$_2$ cladding refractive index. Therefore, there is a considerable loss in this

wavelength region. As shown in Figure 8a, the estimated transmittance under h = 200 nm decreased to less than −3 dB at the wavelength region greater than 1600 nm. Conversely, when we set a higher device height, the TM effective index increased dramatically. In this case, under the previously determined grating periods, the TM polarization state at shorter wavelength region, e.g., <1450 nm, tended towards the Bragg region and suffered from a relative high loss due to the Bragg reflection. As demonstrated in Figure 8a as well, when the device height was set as 240, the transmittance at the wavelength region shorter than 1500 was less than −2 dB. Additionally, if we continue to increase the device height to h = 320 nm, the edge of the TM Bragg window appears at 1430 nm. Hence, as we determined, h = 220 nm is actually a good tradeoff in this proposed polarizer.

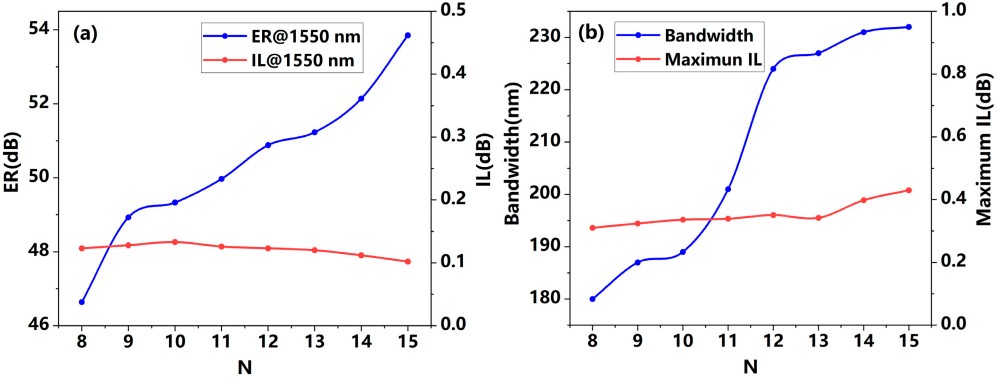

**Figure 7.** (**a**) The extinction ratio and insertion loss as a function of grating period numbers at 1550 nm; (**b**) maximum insertion loss and operating bandwidth as a function of grating period numbers.

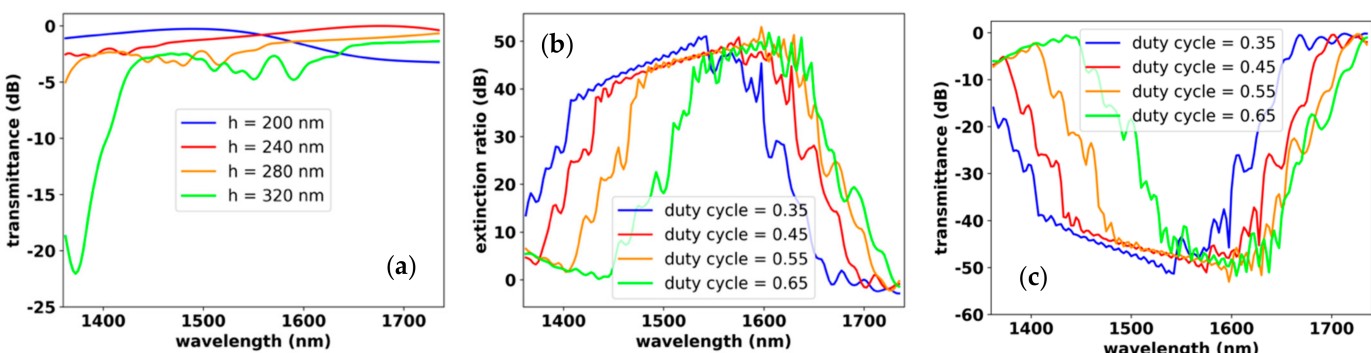

**Figure 8.** (**a**) The wavelength-dependent transmittances of the TM polarization state under various device heights; (**b**) The wavelength-dependent extinction ratio of the proposed polarizer under various duty cycles. (**c**) The wavelength transmittances of the TE polarization state under various duty cycles.

In terms of the duty cycle, we simply increased it from 0.3 to 0.7 so that the grating geometries, especially the grating ridge width, did not challenge the fabrication accuracy. Figure 8b shows the ER calculated using the estimated wavelength-dependent transmittances of TE and TM polarization states under various grating duty cycles. Clearly, the ER window demonstrated a red shift with an increasing duty cycle owing to the shifted TE reflections as demonstrated in Figure 8c. Meanwhile, the bandwidth decreased as the duty cycle increased. When the duty cycle is set to greater than 0.6, the estimated working bandwidth is narrower than 200 nm. However, if the duty cycle is set to less than 0.55, the bandwidth deviation is actually less than 10 nm.

It could be asserted that more grating sections provide a broader polarizer bandwidth, although the IL increases. In this case, we attempted to add two more grating sections with a period of Λ = 300 nm and Λ = 380 nm into the polarizer. Additionally, for each grating section, 10 periods were utilized. Figure 9 shows the evaluated ER and IL by

using the five grating sections. At 1550 nm, the ER is approximately 53.73 dB and the IL is approximately 0.1 dB. Over the wavelength region from 1390 nm to 1700 nm, the ERs are all greater than 20 dB. In other words, the polarizer bandwidth is extended to 310 nm when the five grating sections are utilized. At the same time, the IL of TM polarization state is always smaller than 0.7 dB. In contrast to the three grating sections, the IL increased by less than 0.3 dB. This design has the potential to be utilized in applications with a high demand in bandwidth but low requirement in loss.

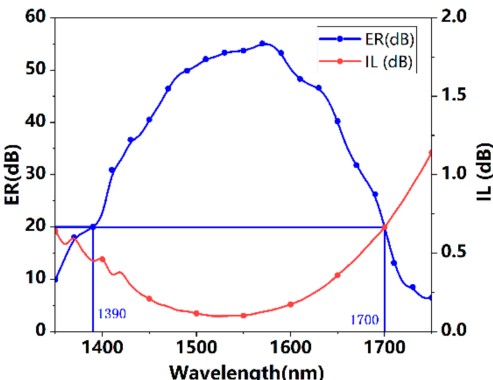

**Figure 9.** Wavelength dependent extinction ratio and insertion loss of the proposed polarizer evaluated under five grating sections.

### 3.2. Device Fabrication Tolerance

Considering that the Bragg reflection is sensitive to the grating period and duty cycle, we performed a statistical analysis to characterize the device manufacturing tolerance [22]. In practice, random jitters are introduced in all grating ridges with a variance of σ = 5 nm, σ = 10 nm, and σ = 20 nm. Additionally, for each kind of jitter variance, ten independent realizations are modeled and simulated using 3D-FDTD calculation. The error bar of the evaluated ER under the three jitter variances are shown in Figure 10a–c. Obviously, for each ridge jitter variance, the ER has higher standard deviation in the wavelength region from 1470 nm to 1630 nm. Additionally, such ER fluctuations become more evident as the ridge jitter variance increases. Nevertheless, even in the worst case, the highest ER is greater than 40 dB and the bandwidth is broader than 200 nm. On the other hand, the IL fluctuation is negligible, so that it is always smaller than 0.5 dB over the wavelength region from 1390 to 1680 nm. Hence, the proposed polarizer is a high fabrication tolerant device.

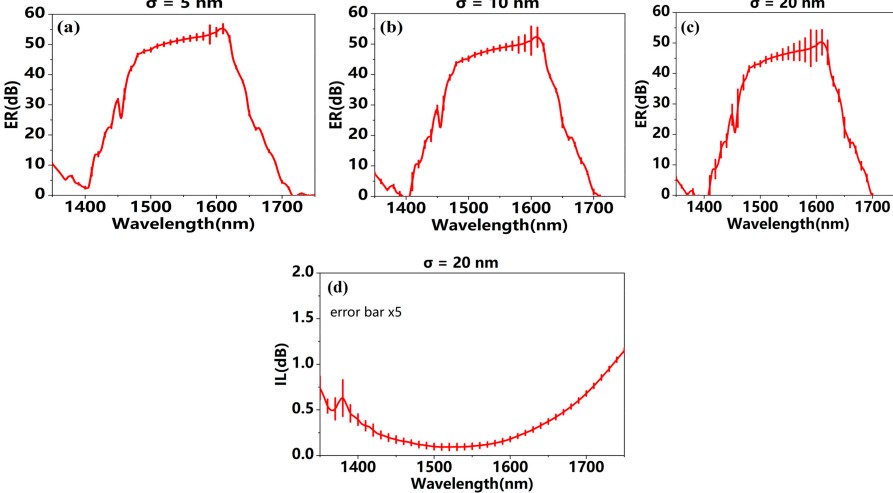

**Figure 10.** Extinction ratio obtained under the grating ridge jitter variance of (**a**) σ = 5 nm; (**b**) σ = 10 nm; and (**c**) σ = 20 nm. (**d**) Insertion loss obtained under the grating ridge jitter σ = 20 nm.

## 4. Conclusions

In conclusion, an ultra-broadband TM-pass/TE-stop polarizer was designed at the communication band by using multistage Bragg grating sections. With a device length of 15.3 μm, the polarizer working bandwidth was extended to 231 nm and its insertion loss was consistently smaller than 0.42 dB over the entire operating wavelength band. If a slightly narrower bandwidth of 225 nm is permitted, a more compact device length of 12.2 μm is achieved. Moreover, to further extend the device bandwidth, two more grating sections are suggested to be added to the polarizer so that the polarizer bandwidth increases to 310 nm, but the insertion loss is also increased to 0.7 dB. Finally, a statistical fabrication tolerance analysis is performed for the proposed polarizer. The smaller ER and IL fluctuation means that the device always has a bandwidth of greater than 200 nm, and even the grating ridge has a 20 nm jitter variance.

**Author Contributions:** Conceptualization, Y.D. and Y.X.; methodology, Y.D.; software, Y.L.; validation, Y.L. and Y.D.; formal analysis, Y.L.; investigation, Y.D. and Y.L.; resources, Y.D. and B.Z.; data curation, Y.L.; writing—original draft preparation, Y.D.; writing—review and editing, Y.D. and Y.L.; visualization, Y.D.; supervision, Y.D.; project administration, Y.D.; funding acquisition, Y.D., Y.X. and B.Z.; All authors have read and agreed to the published version of the manuscript.

**Funding:** This research was funded by the National Natural Science, grant number 6103159, and Natural Science Foundation of Jiangsu Province, grant number BK20190617 and BK20200592.

**Institutional Review Board Statement:** Not applicable.

**Informed Consent Statement:** Not applicable.

**Data Availability Statement:** Data underlying the results presented in this paper are not publicly available at this time but may be obtained from the authors upon reasonable request.

**Conflicts of Interest:** The authors declare no conflict of interest.

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
