# Peer review of "An Ultra-Broadband Design of TM-Pass/TE-Stop Polarizer Based on Multistage Bragg Gratings"

_photonics, doi:10.3390/photonics9060409_

Round 1
Reviewer 1 Report
The manuscript photonics-1738254 reports an TM-Pass/TE-Stop polarizer based on multistage Bragg gratings from a theoretical perspective. The results show that the designed polarizer has a bandwidth up to 321 nm with extinction ratio greater than 20 dB and incision loss less than 0.42 dB. However, there are some issues that the authors might consider.
- As for a theoretical study, the physical principle of mode coupling for the generation of polarization-dependent transmission and hence the TM-pass/TE-stop polarizer should be discussed in depth. For instance, the Bragg grating generally generates a non-polarization-dependent reflectance or transmittance, which is totally different from the case in this study.
- The designed polarizer consists of three Bragg gratings with different periods. The transmission spectra or reflection of these three Bragg gratings should be compared with each other and with that of the designed polarizer, so as to clearly present the generation of a wide band resulted from spectral superposition.
- The mechanism that the resonance band of TE/TM polarization states shifts with width W1 and W2 should be analyzed quantitatively for a theoretical study. For example, why does the resonance wavelength of TM polarization keep stable as W1 and W2 change while TE polarization shift towards longer wavelength?
- How the authors calculated the insertion loss in a theoretical study, a parameter that is greatly dependent on instruments in experiment.
Reviewer 2 Report
The authors presented a study of broadband waveguide polarizer, which is based on multistage Bragg gratings. The results show that the working wavelength range reaches to 231 nm and an extinction ratio greater than 20 dB. This work looks good. However, there are some issues should be addressed.
1. The authors should include a theory and principle part to explain the basic principle of the device exhibiting the TM-pass/TE-Stop polarization.
2. Followed by the prior comment, the authors are suggested to provide mode profile since the device is polarization-related.
3. The authors simulated the device by varying the widths, however, the height of the device is also a factor which could influence the device performance. The authors are suggested to provide such information.
4. The authors show the extinction ratio and insertion loos as a function of grating period number in Figure 7. Why the insertion loss decreases as the grating period number increases?
5. Since this research is focused on broadband polarizer by multistage Bragg gratings. Does the authors consider that whether the period duty cycle varies the band? If possible, the authors are suggested to provide this information.
Reviewer 3 Report
This paper proposed a wideband PBS based on multi-stage waveguide Bragg gratings with various periods. Their simulations show good results, but with no experimental validations. Furthermore, from my point of view, the work lacks sufficient novelty. The so-called "multistage" waveguide Bragg gratings are essentially chirped Bragg gratings, and using such chirped Bragg gratings to extend operation bandwidths is not new. I also have one additional question:
1. Could authors explain why the bandgap for TM mode is much smaller than that for TE mode forsuch Bragg grating structures?
Round 2
Reviewer 1 Report
The authors have addressed all the concerns and improved the manuscript.
Author Response
As indicated by the reviewer, I have already addressed all his comments and suggestions in the previous round of revision.
Reviewer 2 Report
The authors have addressed all of my comments. This revised version looks good and can be accepted.
Author Response
As indicated by the reviewer, I have already addressed all the comments in the previous round of revision.
Reviewer 3 Report
Thanks the authors for the response.
The author's response explained why the TM mode's bandgap is located at a much shorter wavelength than TE. However, the point of why TM bandgap is much narrower than that of TE mode is still not clearly explained.
Perharps the authors meat that the TM modes are much less sensitive to the width variations than TE mode, and so the coupling coefficient is much smaller? If yes, I believe more clear explanations regarding this point needs to be added in the manuscript. Additional simualtions for the explanations would also be very helpful.
An additional question: in Fig. 6(a): why in some wavelengths (such as 1650 nm), the transmission of the cascaded gratings can be even larger than individual gratings (orange curve:period = 360 nm)?
